# Personality and Person-Work Environment Fit: A Study Based on the RIASEC Model

**DOI:** 10.3390/ijerph20010719

**Published:** 2022-12-30

**Authors:** Jonatan Santana Batista, Sônia Maria Guedes Gondim

**Affiliations:** 1Institute of Psychology, Universidade Federal da Bahia, Salvador 40-210730, Brazil; 2Institute of Psychology, Universidade Federal de Uberlândia, Uberlândia 38408-100, Brazil

**Keywords:** congruence, personality, RIASEC, work fit, professional interests

## Abstract

This study identifies if there are differences in the personality scores of professionals with varying degrees of congruence, considering each dimension of the RIASEC model. Method: A cross-sectional survey study. Participants responded to three measures: Vocational Interests Scale (VIS); Occupational Classification Inventory (OCI-R) for estimating congruence; and The Next Big Five Inventory (BFI-2) for estimating personality. Results: Congruence was associated with at least one personality dimension in the Realistic, Investigative, Artistic, and Conventional types. In addition, we identified significant differences between the personality scores of professionals according to the degree of congruence in the Realistic, Investigative, Artistic, and Enterprising types.

## 1. Introduction

The RIASEC model developed by Holland [1], with the Realistic, Investigative, Artistic, Social, Enterprising, and Conventional dimensions, adopts the principle that these six general dimensions represent vocational choices. It also argues that each individual’s personality influences such choices. In addition, differences in individual repertoires result in distinct work configurations and preferences.

In Holland’s [1] conception, the RIASEC and the Big Five (Five Personality Factors) models complement each other. Individuals with specific personality scores mostly represent each professional type described in RIASEC, since these attributes are more effective in their respective areas and tend to be innocuous or even undesirable in others. Based on this, professionals choose activities considering their skills, beliefs, and similarities [2].

Hoff et al. [3], Rúa et al. [4], and Wille and De Fruyt’s [5] studies sought to identify, in the Big Five factor model, the prevalent personality characteristics in professionals classified in each of the six RIASEC dimensions. Among the main results, they identified positive relationships between neuroticism and the Social type; extraversion and the Enterprising type; conscientiousness and the Conventional type; openness to experience and the Social and Artistic types; and negative associations between agreeableness and the Investigative type.

Given the abundant evidence that allows the identification of the prevalence of personality scores for each RIASEC type, this study sought to investigate the personality scores of congruent professionals, that is, those whose vocational interests adjust better to job demands. Would there be differences between the personality scores of professionals whose interests matched or did not match with work demands?

This study contributes to understanding the relationships between the Big Five and RIASEC, helping to accumulate evidence on the prevalent personality scores for each type described in the RIASEC model. In addition, it may generate inputs to support people management practices by identifying whether personality traits differ among individuals with distinct levels of adaption to the work environment.

### 1.1. Relationship between the Big Five and the RIASEC Model

For decades, researchers have studied the relationships between personal and occupational characteristics in order to understand how personality traits can influence the process of professional choice and development [6]. The process of professional choice has two stages, according to Woods et al. [7]. The stages are selection and adjustment. The first occurs through the selection of a general and a broad field. The adaption process occurs through a dynamic agreement between individual and job demands, which operates when an individual’s expectations must align with work demands.

Several studies question the extent to which personality dimensions affect work. Would organizations need to keep an alignment between employees’ desirable personal characteristics for each specific work environment as a necessary arrangement for carrying out their processes? Would these attributes represent a substantial impact for work and changes in intraorganizational networks’ interactions? [8,9].

The hexagonal RIASEC model, with its Realistic, Investigative, Artistic, Social, Enterprising, and Conventional vocational types, shown in Figure 1, was developed by Holland [1] in his theory of vocational personalities and work environments. It assumes that work environments maintain job homogeneity, such that individuals in each group have similar work repertoires, beliefs, and culture, when compared to individuals in different work contexts. Thus, there must be equivalence between an individual’s repertoire and the demands of the work environment to ensure reasonable homogeneity, and thereby a cohesive organizational network and the efficient operation of the organization [10].

The RIASEC model of Figure 1 allows the classification of individuals in different areas of professional activity [1,11]. Each dimension has numerous activities. The similarities between the occupations in the RIASEC correspond to the proximity between the dimensions in the hexagonal model [12].

Through its application, it is possible to distinguish professionals by observing the interest/work demand for each area, based on RIASEC’s prevalent dimension [2]. Table 1 classifies the typical tasks and prevalent personality traits for each dimension of the RIASEC model.

Some studies using the Big Five model have sought to identify prevalent personality scores for each RIASEC dimension. Orkibi’s [13] study on burnout and work commitment identified that professionals with interests classified in RIASEC as Artistic and Social mitigated the negative impact of burnout on professional commitment. Wiernik’s [14] study with professionals classified in the Realistic dimension identified low agreeableness, and extraversion varied according to the type of activity: professionals from the production sector had low scores in extraversion compared to professionals from other sectors.

Golle et al. [15] concluded that professionals with high conscientiousness scores tended to prefer a fast entry into the job market, despite their academic background, and had low interest in the Investigative, Social, and Enterprising types of the RIASEC model. Hoff et al. [3] identified that, among all the types described in RIASEC, the Realistic type has few significant associations with personality dimensions compared to other vocational types, thus pointing to inconclusive results.

When mapping the personality scores of professionals classified according to RIASEC, Wille and De Fruyt [5] found negative relationships between Neuroticism and the Enterprising types, which is probably justified because these types require greater emotional management to face several challenges in the decision-making process. On the other hand, professionals with high neuroticism scores preferred Conventional and Social activities. This may occur because conventional activities are more structured and standardized, and social activities favor cooperation (social support) over competition.

The results of the study showed that professionals classified as Realistic presented high scores in conscientiousness, likely because it is a more structured area with a higher technical demand, where speed and expertise are highly desirable. The dimension openness to experience proved to be associated with the Artistic type, characterized by unstructured activities that require creative solutions and improvisation. In a meta-analysis study, Rúa et al. [4] found positive associations between openness and the Artistic and Investigative types; extraversion and the Social and Enterprising types; and conscientiousness and the Enterprising and Conventional types.

### 1.2. Congruence and Personality

Since work is a core element of human life, researchers have sought variables that could improve the relationship between an individual and the work environment. In Holland’s [1] theory of vocational personalities and work environments, the author defines the person-work fit process as congruence, which corresponds to the level of agreement between the professional’s vocational interests and the organization’s work demands. Therefore, we assume that the higher the congruence, the higher the person-work fit will be, and consequently, the higher the likelihood of staying in the organization, good performance, and job satisfaction [11,12,16,17,18,19].

Wille and De Fruyt [5] analyzed the reciprocal interactions between personality and participants’ careers, classified according to the RIASEC model. They observed that work environments are shaped by the characteristics of the professionals that make up an organization, while job demands progressively shape the beliefs, behaviors, and repertoires of the professionals, leading to gradual changes in their personality scores.

In situations where there is no congruence, there is greater pressure for a change from the professional who will need to shift their repertoire of beliefs, interests, and abilities to keep their space at the workplace [20]. In cases of high congruence, there is no demand for change since the individual attributes mostly correspond to the experiences at the work environment. Here, congruence is understood as a balance between certain configurations of the work environment and those at the individual level. The closer they are, the easier it will be to keep the system working efficiently [5,18].

The need to maintain congruence in some work environments may require changes in individual attributes, with an effort by the professional to adapt and remain congruent. A lack of fit will increase the chances of evasion, with the professional leaving the organization and going toward work environments more compatible with their interests [21,22]. Low congruence can cause negative effects for the organization and for the worker, depriving them from accessing the same benefits and rewards compared to those who enjoy a good person-work environment fit [23].

It is easy to imagine that a highly conscientious individual will have difficulty fitting into a mostly Artistic professional environment that operates based on inspiration, with peaks of creative output. The conscientious profile deals better with previously defined and systematic tasks, congruent with Realistic or Conventional environments [1,16,22]. In this case, there is a mismatch between a loosely structured environment that demands creative solutions and flexibility, and a professional who prioritizes deadlines, order, and inspection [1].

Vocational theories such as that of Holland’s [1] highlight the role of personality dimensions as requirements for congruence, where personal characteristics favor entering a variety of work environments whose peer interactions will be easier to maintain [4]. However, given the lack of studies that allow theoretical advancement on the contribution of personality dimensions to the person–environment fit, the present study sought to identify if there were differences in the personality scores of professionals with varying degrees of congruence, considering each dimension of the RIASEC model. This objective is in line with Wille and De Fruyt’s [5] recommendation to conduct studies for investigating the relationships between personality and congruence.

The study allowed us to identify, for example, if poorly congruent professionals have personality scores different to that of congruent professionals. It also contributes to provide inputs for practitioners and researchers, both in the selection stages and in processes of organizational socialization in order to facilitate a favorable arrangement for poorly congruent professionals to achieve a better fit.

## 2. Materials and Methods

### 2.1. Participants

The sample had 504 participants (284 men, 214 women, and two who did not declare their sex), professionals from various segments classified according to the RIASEC model. The inclusion criteria were (a) had been at the organization for at least six months; (b) participants had to be over 18 years old. The professionals recruited at the time of data collection worked in private Brazilian companies in the state of Bahia. The data collection strategy took place through access to groups of professionals on social networking pages such as Discord and Facebook. The average time in the organization was 5.7 years (SD: 8 years).

In terms of educational level, 5 had completed elementary school; 16 had not finished high school; 62 had completed high school; 123 had incomplete higher education; 139 had an undergraduate degree; and 155 had a graduate degree. As for income, 112 participants earned up to 1.5 minimum wages; 201 between 1.5 and 3 minimum wages; 86 between 3 and 5; 57 between 5 and 10; and 44 earned more than 10 minimum wages. The participants were classified as Realistic (49), Investigative (82), Artistic (13), Social (203), Enterprising (120), and Conventional (33).

### 2.2. Ethical Procedures

We submitted the study proposal to the Ethics Committee for Research in Psychology (CEP), a competent entity that investigates the research conditions and the adequacy of proposals for conducting data collection with human beings (CAAE: 52354321.1.0000.5686). The project was also approved by the coordination of the graduate program in Psychology (PPG-PSI), responsible for authorizing studies in the institution.

### 2.3. Instruments

#### 2.3.1. Vocational Interest Scale (VIS)

The Vocational Interest Scale (VIS) [24] has six RIASEC dimensions, with forty-eight items, eight items per dimension. The scale requires respondents to rate their interest in activities described in the instrument. Participants answer based on a Likert scale ranging from 1 (I dislike it very much) to 5 (I like it very much). The psychometric properties were internal consistency (Cronbach’s alpha): R (0.74), I (0.77), A (0.75), S (0.79), E (0.67), and C (0.71). Principal axis analysis (AEP) with Oblimin rotation showed adequacy of the items to the six-factor model, explaining 34.1% of the total variance.

#### 2.3.2. Occupational Classification Inventory (OCI-R)

OCI-R is the revised version of the Occupational Classification Inventory [25]. The measure has six RIASEC dimensions (Realistic, Investigative, Artistic, Social, Enterprising, and Conventional), with fifty-four items and nine items per dimension. The wording of the items asks the respondent to report how often a person in their position does the activities. Participants mark their answers on a scale ranging from 1 (Never) to 5 (Always). The psychometric properties are internal consistency (Cronbach’s alpha) R (0.73), I (0.85), A (0.84), S (0.86), E (0.81), and C (0.81). Altogether, principal axis analysis (AEP) with Oblimin rotation all six factors explained 40% of the total variance.

#### 2.3.3. The Next Big Five Inventory (BFI-2)

The measure relies on the Big Five Personality Factors model [26] and has seventy-six items distributed in the five dimensions: Neuroticism, Extraversion, Socialization, Conscientiousness, and Openness to Experience. The measure features a Likert scale ranging from 1 (It has everything to do with me) to 5 (It has nothing to do with me). The dimensions correspond to extraversion (α = 0.87), agreeableness (α = 0.82), conscientiousness (α = 0.84), neuroticism (α = 0.86), and openness (α = 0.82). The paper with the national version of the measure is still in progress.

### 2.4. Data Collection

We collected data online using Google’s questionnaire generator (Google forms). We sent the link with the Free and Informed Consent Form (TCLE), response instructions, and the measures together. Participants first agreed to participate in the survey in order to have access to the instruments.

### 2.5. Data Processing

We did the following: (a) assessment and cleaning of the database, where we identified low representation among the participants in the Artistic dimension of RIASEC; (b) psychological measures were inspected, using the Kolmogorov–Smirnov and Shapiro–Wilk tests to check the normality assumptions. Data presented a non-normal distribution, so we decided to carry out nonparametric analyses. After this stage, we created the variables referring to the dimensions of the psychological measures.

### 2.6. Calculation of the Congruence Score and Classification of Participants

The procedures for obtaining the congruence score depended on using the Occupational Classification Inventory (OCI-R), which measures work demands, and the Vocational Interest Scale (VIS), which measures interests. With these instruments, we conducted the following three procedures:(1)The top three interest/environment scores were selected. These scores were tabulated according to the position occupied in the hexagonal model and then compared in the next step, the value being 1 for Realistic, 2 for Investigative, 3 for Artistic, 4 for Social, 5 for Entrepreneur and 6 for Conventional.(2)Using Figure 2 as a reference for the Realistic dimension, we created three new variables by comparing the scores of interests and environments generated in the previous step. When the pair had the same value for interest/environment (identical pairs), we assigned a score of 3. When the interest/environment pair was close, we assigned a value of 2 (adjacent pairs). If the interest/environment pair was far (alternating pairs), we assigned a value of 1. Finally, if interest and environment were in opposite positions (opposite pairs), we assigned a zero (0) value.(3)Then, we used the algorithm C-index [25] that has the equation:
C = 3 (x1) + 2 (x2) + 1 (x3),(1)
where “C” represents congruence and “x1”, “x2” and “x3” represent, respectively. The congruence score of each respective interest/environment pair was obtained in the previous step. The C-index produces a congruence score that varies from 0 (no congruence) to 18 points (maximum congruence).

After achieving the congruence score, participants were divided into two halves, based on the median value. We took this decision because we identified a non-normal distribution. The value 11, corresponding to the median, was used as a parameter so that equal or lower values were classified as “lower congruence”, and higher values classified as “higher congruence”.

### 2.7. Data Analysis

We summarized the analyses in the following procedures: Descriptive statistics of the variables; Separation of the groups of higher and lower congruence; Identification of the personality scores for each RIASEC type; Association between the personality dimensions and congruence for each RIASEC type; Comparison between the personality scores in the groups of higher and lower congruence for each RIASEC type. We used the software IBM SPSS 23.0 (IBM, New York, NY, USA) and R-Studio (Version: 2021.09.0351).

## 3. Results

Table 2 presents the non-parametric descriptive data of personality dimensions for each RIASEC type, using the median and the interquartile ranges. The distribution of the minimum and maximum scores in the dimensions was balanced among all the groups investigated. The low representation of the Artistic type stood out, with only 13 cases identified, followed by the Conventional type, with 33 cases.

Table 3 presents Spearman’s Rho correlation between personality and congruence dimensions for the six RIASEC types. The Realistic type showed weak positive associations with neuroticism (Rho = 0.34; *p* = 0.018), and negative associations with openness (Rho = −0.31; *p* = 0.040). The Investigative type showed weak associations with agreeableness (Rho = 0.22; *p* = 0.043), conscientiousness (Rho = 0.22; *p* = 0.041), and openness (Rho = 0.33; *p* = 0.003). The Artistic type showed strong and negative associations with conscientiousness (Rho = −0.71; *p* = 0.006). Finally, the Conventional type was positively associated with the conscientiousness dimension (Rho = 0.34; *p* = 0.049). We did not identify significant associations for the Social and Enterprising types.

Table 4 presents the results of the Mann–Whitney test for comparing ranks (mean ranks were used to compare dimensions) in the groups with the highest and lowest congruence classified according to the RIASEC. For the Realist type, the most congruent participants had higher scores in neuroticism than the less congruent professionals (U = 189; *p* = 0.025) and lower scores in the openness dimension (U = 158; *p* = 0.004). In the Investigative type, the most congruent professionals had higher scores in agreeableness (U = 462; *p* = 0.004), higher scores in conscientiousness (U = 453; *p* = 0.003) and higher scores in openness (U = 424; *p* = 0.001).

For the Artistic type, the more congruent participants achieved lower scores on the conscientiousness dimension (U = 0; *p* = 0.003). In this case, the U-value, for all comparisons shows that the values of less congruent participants were higher than those for the more congruent. A potential cause is the low representation of the Artistic type in the sample, which may have created a bias.

For the Social type, professionals with greater congruence did not differ from those with less congruence in any of the investigated personality dimensions. For the Entrepreneur type, participants with greater congruence had higher scores than professionals with less congruence only in the extraversion dimension (U = 1288; *p* = 0.011). Finally, for the Conventional type, no significant differences were identified between the scores of professionals.

## 4. Discussion

### 4.1. Distribution of Personality Scores in RIASEC Types

The minimum, maximum, and median values described in Table 2 suggest that the professionals’ scores in the sample were evenly distributed among the measure values. Also, we observed no significant differences among the RIASEC groups regarding the interquartile range. However, some elements are worth mentioning: professionals described as the Investigative type presented high scores in extraversion and agreeableness, although, according to Holland [1], this is a type usually introvert who prefers to act individually. This suggests that the context can cause changes. For example, the academic environment may occasionally require more extraversion and agreeableness. In addition, we did not collect information on the professionals’ working areas, since applied areas require a more extrovert profile than basic science and laboratory activities.

The Social type has individuals with high scores in conscientiousness, which may go in the opposite direction of this type’s preferences, more focused on cooperation and personal and interpersonal development. Some professionals of the Conventional type presented maximum scores in the neuroticism dimension, which is in line with Holland’s theory [1], since the conventional environment, in general, is very structured and stable, requiring a set of predictable and standardized tasks. The difficulty in managing emotional demands makes those who present high scores in neuroticism seek more stable and predictable environments which facilitate routine procedures.

As there were no extreme values in the descriptive analyses which would allow identifying trends among the RIASEC types, we speculated that the distribution of these values was due to the presence of professionals with low and high congruence in the sample. This would be identified if these groups were analyzed separately, which is described in Table 4. In addition, we noticed the association of the scores in the personality dimensions with congruence, in the six RIASEC groups, as described in Table 3, so these data supported the main findings.

### 4.2. Comparison between Personality Dimensions in RIASEC Types

Data presented in Table 3 and Table 4 strengthen the argument that personal characteristics contribute to developing congruence in these professionals, at the work environment. For example, in the Realistic type, congruence was weakly and positively associated with neuroticism and negatively associated with openness. This means that professionals more adjusted to the demands of this environment tend to be more neurotic; that is, they have a higher risk perception and sensitivity to stressful conditions, and less capacity to control emotions. Hoff et al. [3] concluded that the Realistic type was the one with fewer associations with personality dimensions, while Orkibi [13] identified that only the Realistic type was associated with lower scores in the agreeableness dimension.

The Realistic type does manual activities oriented to technical handling and improves themself based on expertise. They are professionals who engage in the same activities for a long period of time. Therefore, this negative association between congruence and the openness dimension in the Realistic type is in line with Holland’s descriptions [1]. Neuroticism, in turn, is linked to the search for more stable environments where they can fit more easily and avoid risks and stress, as is the case of Conventional environments, which share similarities with Realistic ones.

Hence, there seems to be a convergence, since Realistic professionals, when focused only on doing their activities (less open to experience), adapt more easily to work demands, while professionals that are more neurotic have identified Realistic environments as the safest. Realistic activities are manual, as in sports, physical exercise, and in assembly and maintenance. These activities are less susceptible to change, when compared to the Investigative and Artistic types, which involve more technology and innovation.

For the Investigative type, congruence was weakly and positively associated with agreeableness, conscientiousness, and openness (see Table 3). This indicates that these dimensions contribute to the fit of professionals in environments with academic demands that are permeated by analytical demands. We confirmed these data through the comparisons in Table 4, where more congruent professionals stood out precisely in these dimensions.

We found support for the association between conscientiousness and openness, considering that investigative professionals are task-oriented and appreciate results more than interactions with people. They are also professionals with a wide system of beliefs and interests, and constantly need to learn new things and integrate them into their activities, because their field of operation is constantly changing [1,3,4,7].

However, the contribution of agreeableness to the person–environment fit in this type was not yet documented. The literature highlights that these professionals do not prioritize peer relationships and prefer individual activities [1,5]. However, these results may occur in cases where the sample contains numerous professionals who share both Investigative and Social interests, which would be the case for researchers in Human Sciences.

For the Artistic type, Table 3 showed a strong negative association between congruence and Conscientiousness. This indicates that professionals more adapted to artistic demands are not very focused on practical and material results and do not value structured tasks and productivity. These data rely on the comparisons shown in Table 4, where professionals that are more congruent presented lower conscientiousness scores. There is also support from the literature, since Artistic type professionals prefer ambiguous environments oriented toward expression, originality, and freedom [1].

The Artistic type does not act in high-pressure, performance-oriented environments, but is more attracted to creative tasks. These professionals have a broad system of interests, tend to develop intellectual skills, and take more time to show results and make decisions. In addition, the activities demanded by an Artistic environment do not involve standardized tasks with fixed references and require developing new and creative solutions, in the opposite direction of the ordered and standardized tasks typical of the demands from Realistic and Conventional environments.

Professionals with mostly artistic interests are a difficult category to find and standardize. Few Brazilian organizations operate according to these demands, so these professionals outside universities usually work with multiple ties that are more fragile and temporary. Unlike Conventional, Realistic, Enterprising and Social professionals, which are easily found in organizations, artistic types are hardly found with formal and lasting links, so that it is possible to question to what extent the notion of congruence applies to this type.

Given the nature of their activities, artistic professionals usually provide services for several individuals/organizations, so that the correspondence between their interests and job demands is divided among these jobs. Therefore, the concept of congruence may work differently for this professional category; therefore, we suggest new studies that emphasize investigating this RIASEC dimension.

Table 3 shows that, on the contrary, the Conventional type shows weak and positive associations between congruence and conscientiousness. This suggests that they need to emphasize order, results, and meet deadlines to fit these environments. These data confirm Rúa et al. [4] findings, where the Conventional and Enterprising types achieved the highest scores on this dimension. Professionals adapted to these demands do routine and standardized activities. Therefore, meeting deadlines and goals is a necessary condition to ensure employability and job progression.

Table 3 further shows that the Social and Enterprising types showed no association with congruence, therefore signaling that personality dimensions in general do not contribute to work fit in such environments. When the sample was divided by groups of higher and lower congruence (see Table 4), we found that the Enterprising type, in general, had higher scores on the extraversion dimension. This result is in line with the findings of Hoff et al. [3].

Both Social and Enterprising demands involve constant interactions between peers and customers. The Enterprising type, however, shows more aggressive demands, are oriented to persuasion, leadership, and sales. Extrovert professionals are expected to develop the skills needed in this environment more easily and interact with others with a better adjustment compared to professionals with lower extraversion scores.

### 4.3. Theoretical and Practical Implications

From a theoretical point of view, results confirmed the findings of several studies that associated personality dimensions with RIASEC types. We noticed that some personality scores prevail in certain types, and this strengthens the argument that vocational choices are linked to individuals’ personalities. The study also found evidence that congruence is associated with some personality traits, and that higher congruent professionals tend to express them more than their lower congruent peers do.

In addition, the study leads to the conclusion that the person–environment fit involves accommodation, showing that professionals tend to present a capacity to adapt to work demands. That is, human beings use their flexibility to deal with the need to preserve congruence at work, avoiding incongruence that may bring some damage to their career in the organization and directly affect their personal welfare.

From a practical standpoint, companies that select professionals and adopt the assessment of personality scores can use them to understand which characteristics are useful to facilitate the insertion and adjustment of the professional to the organization. Furthermore, they can design practices to encourage the development of some personal attributes that could help their professionals to integrate more quickly into the workforce and incorporate organizational skills, beliefs, and practices more efficiently.

The study data also strengthen the heuristic value of the RIASEC model for classifying occupations and work environments and provide a parameter for comparing personal and occupational characteristics. Human Resource professionals can evaluate jobs using RIASEC and assist students and professionals in their career selection and development.

### 4.4. Limitations

Although the study makes significant progress, we point to some limitations. First, future studies should diversify work environments even more in order to increase the chances that all types are better represented numerically. Second, considering the sampling limits, we could not do analyses using the sex variable, since women mostly occupy work environments classified in RIASEC as Social and Artistic, while the Realistic type is mostly represented by men. Given the importance of sex occupational stereotypes in the person–environment fit, we recommend including this variable in the testing model of future studies. Third, despite the study having assumed the minimum age as an inclusion criterion, the study did not control for the age of the participants, disregarding possible biases about the personality development of younger people (e.g., 18 years old) when compared to professionals with a more advanced age (e.g., 50 years).

Fourth, the study was carried out based on linear bivariate relationships between RIASEC types and personality dimensions. In this way, professionals are classified based on their predominant interests. Then, the construct of congruence and personality are associated. This methodology limits the potential of the findings, as it may not adequately represent the data set corresponding to the profiles of interest, nor the interrelationships between the personality dimensions. Fifth, some of the measures used did not present ideal psychometric parameters in this sample, which may, on the one hand, compromise the findings, and on the other, point to the need for improvement.

## 5. Conclusions

This study confirmed some findings of the literature that compared the personality dimensions of the Big Five model with RIASEC typology. The major contribution of the study was to identify that some of the characteristics that can contribute to the person–environment fit process, assisting professionals in adapting to work demands.

The next step is to expand the study sample, seek representation of all RIASEC types, and control for variables such as working time and sex. In addition to some jobs that are preferentially occupied by women, and others by men, there is evidence that congruence tends to increase proportionally with working time.

The study data identified relationships between congruence and personality in RIASEC types. Researchers should carry out further local studies to identify the benefits of congruence for the individual, for work teams, and for the organization. With this, it will be possible to increase interest in this topic and encourage both researchers and professionals to join the RIASEC model and incorporate it into their practices.

## Figures and Tables

**Figure 1 ijerph-20-00719-f001:**
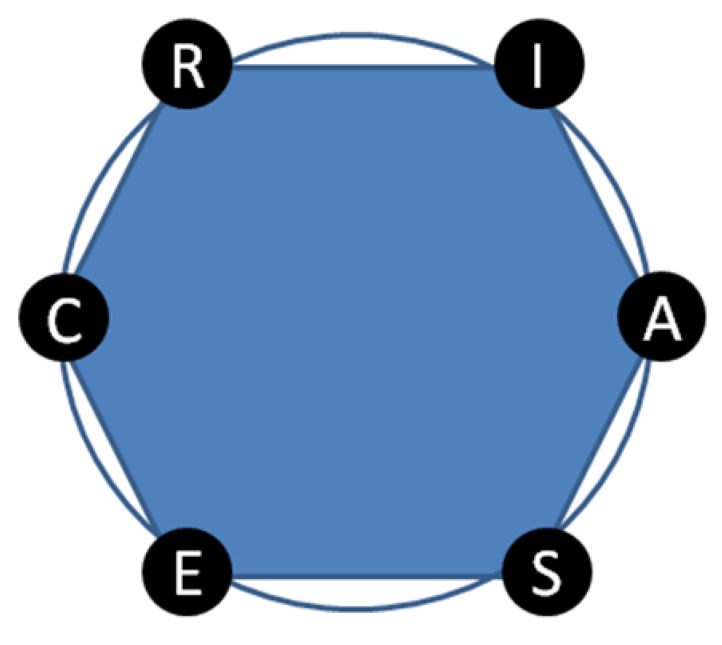
RIASEC hexagonal model [1].

**Figure 2 ijerph-20-00719-f002:**
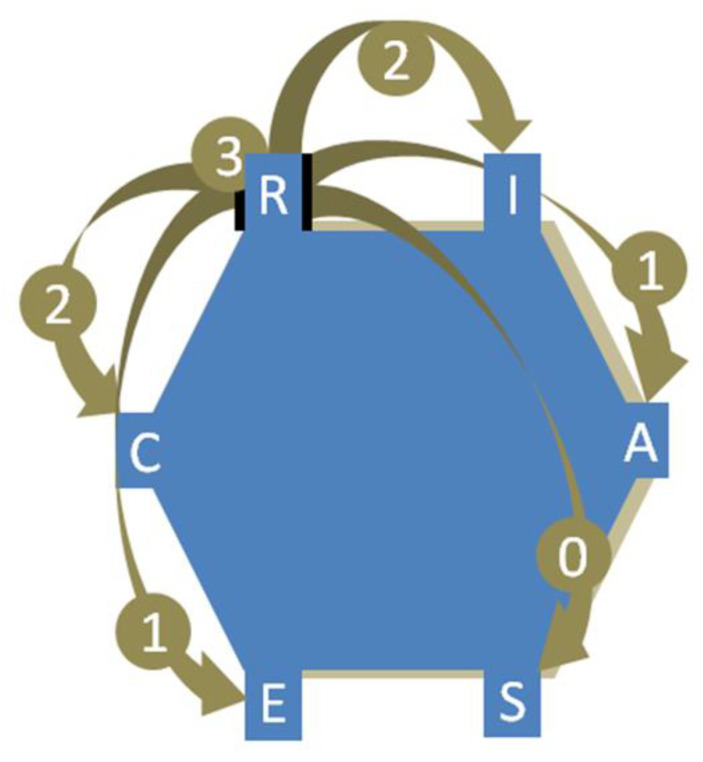
RIASEC hexagonal model and levels of congruence for the Realistic dimension.

**Table 1 ijerph-20-00719-t001:** Brief description of Personality Tasks and Characteristics in the RIASEC Model.

RIASEC	Tasks	Traits
Realistic	Manual tasks involving object manipulation	Conscientiousness
Investigative	Analysis tasks oriented to understanding and explaining phenomena	Introversion, Openness
Artistic	Tasks oriented to innovative and flexible production/expression.	Openness
Social	Tasks that emphasize interpersonal support and development	Extraversion, Neuroticism
Enterprising	Tasks involving leadership, with financial and organizational emphasis	Extraversion, Conscientiousness
Conventional	Tasks focused on control, planning, and standardization.	Neuroticism, Conscientiousness
Source: [4,5]		

**Table 2 ijerph-20-00719-t002:** Descriptive data of personality dimensions for each RIASEC type.

Personality	RIASEC								
	Realistic			Investigative			Artistic		
	Min	Max	Med	Min	Max	Med	Min	Max	Med
Extraversion	1.7	4.7	3.0 (0.82)	1.8	5.0	3.2 (0.86)	2.3	3.3	2.8 (0.64)
Agreeableness	2.2	4.6	3.3 (0.79)	2.3	4.6	3.4 (0.96)	2.6	4.5	3.2 (0.60)
Conscientiousness	1.8	4.7	3.3 (0.79)	2.2	4.8	3.4 (0.93)	2.6	4.6	3.2 (0.82)
Neuroticism	1.6	4.8	2.8 (0.81)	1.3	4.2	2.9 (0.80)	1.5	4.2	3.1 (0.81)
Openness	2.1	4.5	2.9 (0.82)	2.3	4.7	3.6 (0.86)	2.4	4.0	3.2 (0.86)
N	49			82			13		
Personality	**Social**			**Enterprising**			**Conventional**		
	Min	Max	Med	Min	Max	Med	Min	Max	Med
Extraversion	2.1	4.8	3.3 (0.71)	1.6	4.8	3.2 (0.61)	2.5	4.8	3.1 (0.64)
Agreeableness	2.1	4.8	3.4 (0.86)	1.8	4.6	3.3 (0.62)	2.4	4.5	3.4 (0.98)
Conscientiousness	2.1	5.0	3.3 (0.86)	2.4	4.9	3.3 (0.57)	2.6	5.0	3.5 (0.93)
Neuroticism	1.0	4.6	2.7 (0.69)	1.2	4.3	2.9 (0.62)	1.4	5.0	2.7 (1.04)
Openness	2.1	4.7	3.3 (0.93)	2.1	4.3	3.1 (0.70)	2.1	4.3	3.0 (0.39)
N	203			120			33		

**Table 3 ijerph-20-00719-t003:** Spearman’s correlation between Congruence and Personality in RIASEC types.

Realistic					Investigative				
Ext	Agr	Cons	Neu	Opn	Ext	Agr	Cons	Neu	Opn
−0.04	−0.08	−0.05	0.34 *	−0.31 *	0.13	0.22 *	0.22 *	−0.14	0.33 **
**Artistic**					**Social**				
Ext	Agr	Cons	Neu	Opn	Ext	Agr	Cons	Neu	Opn
−0.28	−0.33	−0.71 **	0.03	−0.02	0.13	0.07	0.11	−0.08	0.12
**Enterprising**					**Conventional**				
Ext	Agr	Cons	Neu	Opn	Ext	Agr	Cons	Neu	Opn
0.17	0.00	0.11	0.00	0.00	0.21	0.32	0.34 *	0.19	0.32

Note. Ext: Extraversion; Agr: Agreeableness; Cons: Conscientiousness; Neu: Neuroticism; Opn: Openness. * *p* < 0.05; ***p* < 0.01

**Table 4 ijerph-20-00719-t004:** Test for comparing ranks between groups of higher and lower congruence.

V	**Congruence**		**Statistics**		**Congruence**		**Statistics**	
	Lower	Higher	U	*p*	Lower	Higher	U	*p*
	**Realistic**				**Investigative**			
Ext	24.3	25.6	285	0.756	37.9	48.3	566	0.062
Agr	28.5	21.5	215	0.087	36.1	52.1	462	0.004
Cons	26.8	23.2	256	0.373	35.9	52.3	453	0.003
Neu	20.3	29.4	189	0.025	43.5	37.6	647	0.284
Opn	30.9	19.3	158	0.004	35.3	53.4	424	0.001
	**Artistic**				**Social**			
Ext	7.7	6.3	17	0.517	97.2	105.2	4571	0.342
Agr	7.7	6.3	17	0.518	100.8	102.7	4868	0.821
Cons	10.5	4.1	00	0.003	102.7	101.5	4899	0.880
Neu	6.5	7.4	18	0.667	101.6	102.2	4933	0.946
Opn	6.8	7.1	20	0.886	93.3	107.8	4254	0.085
	**Enterprising**				**Conventional**			
Ext	53.4	69.7	1288	0.011	16.2	17.4	130	0.811
Agr	60.5	60.5	1767	0.988	14.8	19.3	99	0.182
Cons	56.6	65.3	1501	0.157	14.4	19.7	93	0.121
Neu	60.7	60.2	1754	0.943	15.2	18.9	105	0.270
Opn	60.1	61.1	1736	0.865	15.7	18.4	114	0.426

Note: V = Variables; Cong = Congruence; Ext = Extraversion; Agr = Agreeableness; Cons = Conscientiousness; Neu = Neuroticism; Opn = Openness. U = Mann-Whitney; TDE-LC = Size of the effect in common language. Lower = Average rank of the group with lower congruence; Higher = Average rank of the group with higher congruence.

## Data Availability

The database, table and figures are fully available at the time of submission.

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
