# Peer review of "Personality and Person-Work Environment Fit: A Study Based on the RIASEC Model"

_ijerph, 2022, doi:10.3390/ijerph20010719_

Round 1

Reviewer 1 Report

Comments and Suggestions for Authors

This study contributes to understanding the relationships between the Big Five and RIASEC models, helping to accumulate evidence on the prevalent personality scores for each type described in the RIASEC model, generating inputs to support people management practices, by identifying if personality traits differ among individuals with distinct levels of adaptation to the work environment.

A minor revision is suggested for this paper.

Overall, this is a well-written paper with interesting results in this domain.

Comments/requests for clarification

Aspects to improve in the article

1. On line 46: "adaption" or adaptation?

2. Line 156: In the selection criteria, participants had to be over 18 years old.

In terms of personality, don't you consider that, in terms of self-assessment, there may be biases when we questioned, for example, workers aged 18 and workers aged 55?

3. How did you evaluate the criterion "participation in a work team" and why was it so important for your investigation?

4. You do not mention who the participants are (only that they are workers) or how the participants were recruited. At a certain point in the article, it seems that they are academy workers, but this is not clear.

5. Please check final references.

When I downloaded the article to review there is a blank space at number 12

6. For future studies:

In addition to carrying out your study in organizations from different sectors of activity, as you refer, I suggest that in terms of method, consider, as a way of overcoming the bias of social desirability, that in terms of filling out the personality and RIASEC questionnaires, another or other members of the organization where the worker performs functions can also fill in - inter-judge agreement. Here, it would make perfect sense for people to carry out their work as a team, so that someone they work with and who knows them well can assess and help analyze the person-work fit process (congruence).

Reviewer 2 Report

The manuscript presents an important contribution to the study of individual differences and person-environment interaction. Below are some suggestions that aim to improve the quality of research and publication.

 The objective of the study is presented in the abstract, namely: "To analyze differences in the personality scores of professionals, considering the degree of congruence.” We know how difficult it is to write the abstract considering the limitation of the number of words. However, the objective itself must be comprehensible to the reader. I emphasize that the way the abstract is written, the reader cannot know what kind of crogruence is being considered. The lines lines 144-148 of the manuscript the objective seems to be more understandably written.

 The hexagonal RIASEC model is presented in Figure 1. This model allows classifying individuals in different areas of professional activity. However, the model’s description is not clear enough. For example, being it a hexagonal model, activities tend to be associated as the dimensions are close in the hexagon, and tend to be minimal or even zero for the more distant dimensions.

 In the description of the subjects participating in the research (lines 158-159) it is noted that "284 men, 214 women, and two who did not declare their gender" participated in the research. Here, the concepts of sex and gender are clearly mixed. The same happens in the discussion section.

 In the participants section, it is necessary to specify in which geographic location the data collection was carried out. Lines 364-365 mention Brazilian organizations. On lines 440-441 it is signed: "Brazilian researchers should carry out further local studies to identify the benefits of congruence for the individual, for work teams, and for the organization. Does this suggestion apply only to Brazilian researchers?

In the instruments section, the original psychometric properties of the instruments are presented. However, I consider that new evidence of validity should be presented for the sample of the present study. If it is not presented, the authors must justify this absence (AERA et al, 2014; Andrade et al., 2018).

 As indicated by the APA, the tables must be presented in the text and then discussed. In the current version of the article, the authors discuss the tables before presenting them to readers.

In lines 421-423 is written:

“We collected data during the SARS-CoV-2 pandemic, when many organizations were adopting the home-office model. Maybe this new challenge has influenced the answers from the participating professionals.”

It is important to indicate to readers what kind of respose bias was caused by home-office model. Do the authors suggest any hyphotesis?

In lines 219-220 it is indicated that “The three main scores of EIV and ICO were converted into the letters that represent each dimension of the RIASEC model. It is necessary to explain how the three scores are calculated if there are, in fact, six of them according to the instruments section.

The methodological limitations are presented in the end of manuscrip. However, the Holand’s model limitation is not discussed in the manuscript. For example, Armstrong et al. (2008) argued that “An important limitation of research linking personality and abilities to the RIASEC types is that the focus in most studies has been on linear bivariate relationships. This methodology fails to capture the multidimensional nature of interest structure and may not effectively represent the interrelations between individualdifferences domains”.

Format settings:

There are wrong abbreviations in the instruments section. e.g.: Vocational Interests Scale (EIV); Occupational Classification Inventory (ICO).

It is necessary to adjust the initials "Abt" to "Openness".

The data analysis section is presented in topics. The topics cannot be written in points according to the rules of the A.P.A.

Standardize the zero before the point of the result values in fractions. Sometimes zero is displayed, sometimes it is not (A.P.A. default).

Sometimes the initials of the big five factors and dimensions of the RISEAC model are capitalized, sometimes they are not. Standardize.

Lines 40-41.

Where it is written:

“So, would there be differences between the personality scores of professionals whose interests matched or not work demands?”

Suggested fit for:

“So, would there be differences between the personality scores of professionals whose interests matched or not WITH work demands?

In line 185 (2.3 instruments) the abbreviation “ICO-R” is presented, in another part of the text “ICO” was used.

In lines 270-272 is indicated: “In the Investiga-270 tive type, the most congruent professionals had higher scores in Pleasantness (U = 462; p 271 = .004)... [..]. The term "Pleasantness" has not appeared before.

Lines 326-327

Where it is written:

“Neuroticism, in its turn, is linked to the search for more stable environments [..].

Suggested fit for:

“Neuroticism, IN TURN/ ITSELF, is linked to the search for more stable environments [..].

Lines 403-404

Where it is written:

“That is, human beings use their flexibility to deal with the need for keeping congruence at work, avoiding incongruence that may bring some damage to their career [..].

Suggested fit for:

“That is, human beings use their flexibility to deal with the need for PRESERVING congruence at work, avoiding incongruence that may bring some damage to their career [..].

Line 406

Where it is written:

“From a practical standpoint, companies that select professionals and adopte the assessment of personality [..].

Suggested fit for:

“From a practical standpoint, companies that select professionals and ADOPT the assessment of personality [..].

References

 American Educational Research Association [AERA], American Psychological Association [APA], and National Council on Measurement in Education [NCME] (2014). Standards for educational and psychological testing. American Educational Research Association.

Armstrong. P., Day, S. X., & Rounds, J. (2008). Holland's RIASEC model as an integrative framework for individual differences. Journal of Counseling Psychology, 55(1), 1–18. 10. https://doi.org/1037/0022-0167.55.1.1

Andrade, J. M., & Valentini, F. (2018). Guidelines for the Construction of Psychological Tests: Regulation CFP No: 009/2018 in Highlight. Psicologia: Ciência e Profissão, 38(núm. esp.), 28-39. https://doi.org/10.1590/1982-3703000208890
